# On-Chip Integrated Yb^3+^-Doped Waveguide Amplifiers on Thin Film Lithium Niobate

**DOI:** 10.3390/mi13060865

**Published:** 2022-05-30

**Authors:** Zhihao Zhang, Zhiwei Fang, Junxia Zhou, Youting Liang, Yuan Zhou, Zhe Wang, Jian Liu, Ting Huang, Rui Bao, Jianping Yu, Haisu Zhang, Min Wang, Ya Cheng

**Affiliations:** 1State Key Laboratory of High Field Laser Physics and CAS Center for Excellence in Ultra-Intense Laser Science, Shanghai Institute of Optics and Fine Mechanics (SIOM), Chinese Academy of Sciences (CAS), Shanghai 201800, China; zhangzhh4@shanghaitech.edu.cn (Z.Z.); zhouyuan00061@163.com (Y.Z.); wangzhe@shanghaitech.edu.cn (Z.W.); yujp_ecnu@163.com (J.Y.); 2Center of Materials Science and Optoelectronics Engineering, University of Chinese Academy of Sciences, Beijing 100049, China; 3School of Physical Science and Technology, ShanghaiTech University, Shanghai 200031, China; 4The Extreme Optoelectromechanics Laboratory (XXL), School of Physics and Electronic Science, East China Normal University, Shanghai 200241, China; 52180920026@stu.ecnu.edu.cn (J.Z.); 15253172638@163.com (Y.L.); 52210920027@stu.ecnu.edu.cn (J.L.); ht19930312@163.com (T.H.); 17860550393@163.com (R.B.); hszhang@phy.ecnu.edu.cn (H.Z.); mwang@phy.ecnu.edu.cn (M.W.); 5State Key Laboratory of Precision Spectroscopy, East China Normal University, Shanghai 200062, China; 6Collaborative Innovation Center of Extreme Optics, Shanxi University, Taiyuan 030006, China; 7Collaborative Innovation Center of Light Manipulations and Applications, Shandong Normal University, Jinan 250358, China; 8Shanghai Research Center for Quantum Sciences, Shanghai 201315, China

**Keywords:** lithium niobate, Yb^3+^-doped waveguide amplifier, low propagation loss, internal net gain

## Abstract

We report the fabrication and optical characterization of Yb^3+^-doped waveguide amplifiers (YDWA) on the thin film lithium niobate fabricated by photolithography assisted chemo-mechanical etching. The fabricated Yb^3+^-doped lithium niobate waveguides demonstrates low propagation loss of 0.13 dB/cm at 1030 nm and 0.1 dB/cm at 1060 nm. The internal net gain of 5 dB at 1030 nm and 8 dB at 1060 nm are measured on a 4.0 cm long waveguide pumped by 976 nm laser diodes, indicating the gain per unit length of 1.25 dB/cm at 1030 nm and 2 dB/cm at 1060 nm, respectively. The integrated Yb^3+^-doped lithium niobate waveguide amplifiers will benefit the development of a powerful gain platform and are expected to contribute to the high-density integration of thin film lithium niobate based photonic chip.

## 1. Introduction

Featuring high-efficiency and broad-gain bandwidth, the ytterbium ion (Yb^3+^)-doped laser and fiber amplifiers operating near the 1 micrometer (μm) wavelength have been extraordinarily successful in industrial processing of materials, as such devices can be operated with extremely high output power [1,2]. In addition to bulk solids and fiber, the Yb^3+^-doped integrated waveguide lasers and amplifiers have become a new technology for on-chip applications. On-chip Yb^3+^-doped waveguide lasers and amplifiers have been previously demonstrated on various substrates including glasses, tantalum pentoxide (Ta_2_O_5_), aluminum oxide (Al_2_O_3_), yttrium-aluminum-garnet (YAG), calcium fluoride (CaF_2_), potassium yttrium tungstate (KY(WO_4_)_2_), and yttrium lithium fluoride (LiYF_4_) [3,4,5,6,7,8,9,10,11,12,13,14]. However, these material platforms cannot simultaneously support ultralow propagation loss, highly-efficient optical nonlinearities and ultrafast electro-optical modulation. Lithium niobate (LN) provides an attractive option as an alternative host material for rare earth ions, besides their broad optical transparency window, low optical loss, high refractive index, high nonlinear coefficient, and large electro-optical effect [15]. It has been demonstrated integrated in high-power lasers on a TFLN platform [16,17]. The on-chip Yb^3+^-doped waveguides have already been demonstrated on weakly guiding ion-diffused waveguides in Yb^3+^-doped bulk LN [18,19,20]. These waveguides feature weak optical confinement and large bending radius due to the low index contrast. Therefore, they are unsuitable for dense integration and also challenging to realize high-gain waveguide amplifiers. In the last two decades, the rapid developments of the wafer-scale, high-quality, thin film lithium niobate on insulator (TFLNOI) and micro- and nano-fabrication techniques have realized on-chip integrated TFLNOI photonic devices with unprecedented performance [21,22]. The on-chip microlasers and amplifiers based on the Er^3+^-doped TFLNOI were demonstrated recently, showing great promise for high-performance scalable light sources based on integrated photonics [23,24,25,26,27,28,29,30]. The on-chip Yb^3+^-doped microlasers based on microresonator have also been demonstrated on the TFLNOI recently [31,32]. Yet to date, on-chip Yb^3+^-doped amplifiers have not been demonstrated on TFLNOI.

Here, for the first time we demonstrate the on-chip Yb^3+^-doped lithium niobate waveguide amplifiers fabricated by photolithography-assisted chemo-mechanical etching. The fabricated Yb^3+^-doped lithium niobate waveguides demonstrate low propagation losses of 0.13 dB/cm at 1030 nm and 0.1 dB/cm at 1060 nm based on the cut-back measurements of different lengths of waveguides. The internal net gains of 5 dB at 1030 nm and 8 dB at 1060 nm are demonstrated on a 4.0 cm long waveguide pumped by 976 nm laser diodes which indicates the gain per unit length are 1.25 dB/cm at 1030 nm and 2 dB/cm at 1060 nm. The integrated Yb^3+^-doped lithium niobate waveguide amplifiers will benefit the development of a powerful gain platform and are expected to contribute to the high-density integration of TFLNOI-based photonic chip.

## 2. Materials and Methods

The on-chip integrated Yb^3+^-doped LN waveguides are fabricated on a 600-nm-thickness Z-cut TFLNOI (NANOLN, Jinan Jingzheng Electronics Co., Ltd., Jinan Province, China) with Yb^3+^ concentration of 0.5 mol% using photolithography-assisted chemo-mechanical etching. The structure of ytterbium-doped lithium niobate wafer is as follows: from top to bottom is a 500 nm thick Cr layer, 600 nm thick ytterbium-doped LN layer, 2 μm silicon dioxide layer, and 0.5 mm thick silica layer. The process flow is as follows: First, a femtosecond laser was used to ablate the Cr layer to process the waveguide pattern of Cr mask. Second, chemical mechanical polishing was used to etch the LNOI layer to transfer the waveguide pattern of the Cr mask to the LN film, then the Cr mask was removed above the waveguide by chemical wet etching and chemical mechanical polishing was performed again. Finally, the waveguide after cleaning was ready for testing. More details of the LN waveguides fabrication can be found in our previous work [33]. The refractive index of LN is ~2.2 and the refractive index of SiO_2_ substrate is ~1.45. 

As shown in Figure 1a, the fabricated Yb^3+^-doped waveguide with the length of 4 cm is folded in a spiral layout for dense integration. Figure 1b shows the optical photograph of the Yb^3+^-doped TFLNOI chip with different lengths of waveguides. Figure 1c shows the scanning electron microscope (SEM) image of the cross section of the fabricated Yb^3+^-doped LN waveguide with a top-width of ~1.1 μm and a bottom-width of ~4.9 μm. The LN waveguide is coated with a thin layer of gold film for the sake of clear imaging in the SEM process, thus both the top surface and sidewall appear a little bit rough. The selection of the waveguide width is to make it as narrow as possible, in order that the single-mode transmission is mainly used in the waveguide. Figure 2a shows the energy level diagram of Yb^3+^, which is the basis of achieving the amplification at different signal light wavelengths [34]. Figure 2b,c shows situ quantitative elements analysis of fabricated Yb^3+^-doped wafer measured by energy dispersive spectrometer (EDS) (Zeiss Gemini SEM450, Carl Zeiss AG, Oberkochen, Germany). As shown in Figure 2b, ytterbium element is detected. The map analysis by EDS in Figure 2c corroborates the uniform micro-scale distribution of ytterbium element in waveguide samples. This provides a favorable factor for the waveguide amplifier to achieve uniform amplification over the entire transmission length.

The experimental setup to characterize optical amplification performance of YDWA is illustrated in Figure 3a. Here, a pump laser at 976 nm is provided by a single frequency laser diode (CM97-1000-76PM, II-VI Laser Inc., Singapore), while single frequency laser diodes at wavelength 1030 nm and 1060 nm (II-VI Laser Inc.) are used as the signal light. The polarization states of both the pump and signal laser beam are set to the TE modes by using in-line fiber polarization controllers (FPC561, Thorlabs Inc., Newton, NJ, USA) to achieve the best amplification performance. The pump and signal laser beam are combined and separated by the fiber-based wavelength division multiplexers (WDM) at the input port of the on-chip integrated Yb^3+^-doped waveguide with lensed fibers. The insets of Figure 3a display the Yb^3+^-doped waveguide butt-coupled by lensed fibers. The waveguide facets are polished with cerium oxide powder with a diameter of 1 µm, and the waveguide facets have the same surface smoothness, thus the coupling loss of the different waveguides is basically the same. Both the output light signals are analyzed by a spectrometer (NOVA, Shanghai Ideaoptics Corp., Ltd, Shanghai, China). The powers of the input and output pump and signal laser are measured by a power meter (PM100D, Thorlabs Inc.). The output of the waveguide is also a zoomed in imaging by an objective onto an infrared camera (InGaAs Camera C12741-03, Hamamatsu Photonics Co., Ltd., Shizuoka, Japan) for the observation output mode profiles, and the captured infrared images are further processed by MATLAB for clear display. As shown in Figure 3b–d, both the pump laser and signal laser are in the fundamental mode of the Yb^3+^-doped waveguide.

## 3. Results and Discussion

Before characterizing the gain of the Yb^3+^-doped LN amplifier, the loss of the waveguide amplifier needs to be determined first. We use the cut-back method to measure the loss of the pump light and the signal light in the waveguide. The propagation loss of the waveguide as $\alpha_{l}$ includes the absorption loss induced by ground-state Yb^3+^ ions and the scattering loss induced by the surface roughness of waveguide. Setting $\alpha_{0}$ as the fiber-to-chip coupling losses per facet, and *L* as the length of Yb^3+^-doped LN waveguide amplifier, the transmittance of the waveguide can be expressed as $T\left(L\right)=-\alpha_{l}L-2\alpha_{0}$. The waveguide transmittance is the ratio of the power of the pump light (signal light) input into the lens fiber over the output power from the lens fiber. The loss curves shown in Figure 4 are fitted by measuring the transmittance of waveguides with different lengths of 0.8 cm, 1 cm, 2 cm, 3 cm, and 4 cm. The number of bends is the same in the waveguides to ensure consistent bend loss and the bending radius is 800 nm. Figure 4a–c shows the loss curves of 976 nm pump light, 1030 nm signal light, and 1060 nm signal light, respectively. Through linear fitting, it can be deduced that the propagation loss of the pump light at 976 nm is 6.78 dB/cm, while the propagation losses of the signal light are 0.13 dB/cm at 1030 nm and 0.1 dB/cm at 1060 nm, respectively. Obviously, the transmission loss depends on different wavelengths mainly because Yb^3+^ ions have different absorption coefficients for different wavelengths [35]. The coupling loss of the pump light is 20 dB per facet, while the signal light is 13 dB per facet at 1030 nm and 9 dB per facet at 1060 nm, respectively. The main reason for the high coupling loss is due to the mismatch between the spot size of the waveguide and the lensed fiber (~2 μm diameter) and the lensed fiber is multi-mode at these wavelengths of 976 nm, 1030 nm, and 1060 nm.

Then we measured the internal net gain in a 4.0 cm long Yb^3+^-doped LN waveguide amplifier. The internal net gain is measured by the signal-enhancement method. It was defined by the following equation: $$g=10\mathrm{lg}\frac{P_{\mathrm{on}}}{P_{\mathrm{off}}}-\alpha_{l}L$$
where $P_{\mathrm{on}}$ and $P_{\mathrm{off}}$ are the output powers of the signal light with and without the excitation of the pump light, respectively, and $\alpha_{l}$ is the optical propagation loss per unit length. Figure 5a,b presents the measured signal spectra at 1030 nm and 1060 nm with the increasing pump powers, which apparently shows the signal enhancement. The full width at half maximum of two narrow signal peaks are about 1.5 nm, it follows the linewidth of laser diode. Figure 5c shows the net gain as a function of the increased pump power with fixed signal powers at 1030 nm. The waveguide amplifier optical gain increases rapidly at the small pump powers, and tends to saturate with relatively high pump powers (>13 mW). The maximum internal net gain of ~5 dB is achieved. Subsequently, the wavelength of the signal light changed to 1060 nm, the net gain curve is shown in Figure 5d. A similar signal amplification phenomenon can be seen, the maximum internal net gain of ~8 dB is achieved with the pump power at 48 mW, which is higher than 1030 nm. The conversion efficiency is about 5% (signal power increment divided by pump power). The difference in saturation power between 1030 and 1060 nm is related to the absorption and emission cross sections of different wavelengths in ytterbium ions.

As shown in Figure 6a,b, when the signal light gain tends to saturated, a significant second harmonic generation (SHG) (~488 nm) from the pump light at 976 nm appeared. The exact power of the pump laser of 976 nm is about 14 mW for 1030 nm and 45 mW for 1060 nm, respectively. It can be seen with the naked eye on the surface of waveguide amplifiers, as shown in the right inset in Figure 3a. Additionally, the peaks at 508 nm and 501 nm correspond to the sum frequency generations (SFG) from the pump light and signal light at 1060 nm and 1030 nm, respectively. The SHG and SFG is harmful to the gain of waveguide amplifiers. Therefore, in order to achieve high power amplification, the nonlinear process need to be suppressed.

## 4. Conclusions

In conclusion, we have demonstrated the fabrication and optical characterization of YDWA on the thin film lithium niobate fabricated by photolithography assisted chemo-mechanical etching. The fabricated Yb^3+^-doped lithium niobate waveguides demonstrate low propagation loss of 0.13 dB/cm at 1030 nm and 0.1 dB/cm at 1060 nm. The internal net gain of 5 dB at 1030 nm and 8 dB at 1060 nm are demonstrated on a 4.0 cm long waveguide pumped by 976 nm laser diodes which indicates that the gain per unit lengths are 1.25 dB/cm at 1030 nm and 2 dB/cm at 1060 nm. Further optimizations concerning the Yb^3+^ ions doping concentration to increase the absorption for pump light, using a mode size converter to improve the coupling efficiency and the waveguide geometric design to allow for high-power amplifications, are anticipated to increase the waveguide gain to even higher values, showing the potential of the on-chip applications for photonic integrated circuit (PIC).

## Figures and Tables

**Figure 1 micromachines-13-00865-f001:**
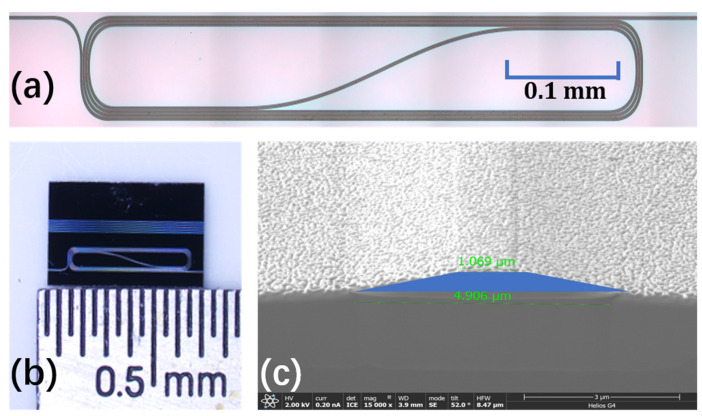
(**a**) Optical micrograph of the fabricated Yb^3+^-doped spiral waveguide. (**b**) The photograph of the Yb^3+^-doped TFLNOI chip with different lengths of waveguides. (**c**) The false color scanning electron microscope (SEM) images of the cross section of the fabricated Yb^3+^-doped waveguide (blue).

**Figure 2 micromachines-13-00865-f002:**
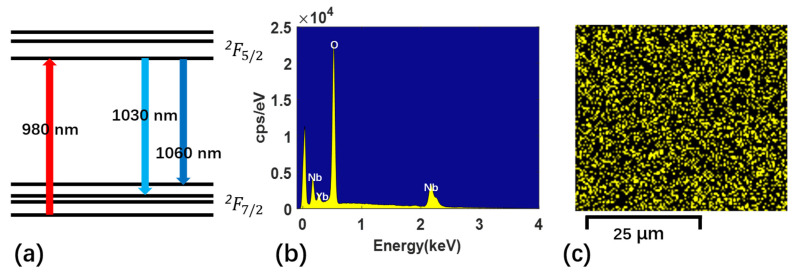
(**a**) The energy level diagram of Yb^3+^. (**b**) The in situ quantitative elements analysis of the fabricated YDWA measured by energy dispersive spectrometer (EDS). (**c**) The map of ytterbium element analysis in waveguide samples by of EDS.

**Figure 3 micromachines-13-00865-f003:**
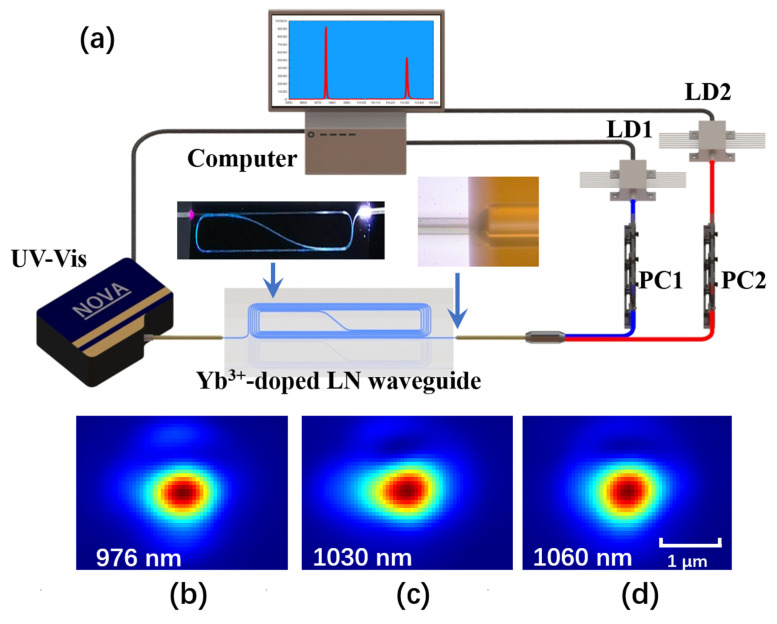
(**a**) The experimental setup to characterize optical amplification of YDWA. Insets show the photograph of the YDWA (left) and the coupling lensed fiber (right). The infrared images of the output port of the fabricated Yb^3+^-doped waveguide with wavelength (**b**) 976 nm laser beam, (**c**) 1030 nm laser beam, and (**d**) 1060 nm laser beam. (LD: laser diodes; PC: polarization controllers; UV-Vis: ultraviolet-visible spectrometer.)

**Figure 4 micromachines-13-00865-f004:**
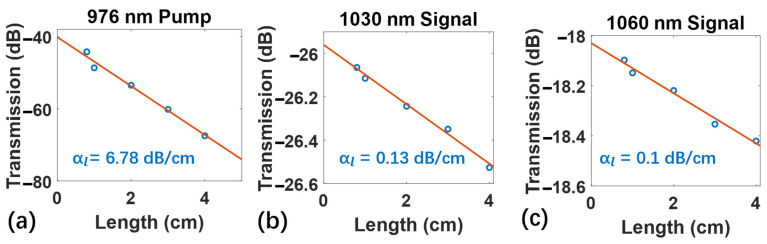
The measured loss curve of the pump light at (**a**) 976 nm, the signal light at (**b**) 1030 nm, and (**c**) 1060 nm.

**Figure 5 micromachines-13-00865-f005:**
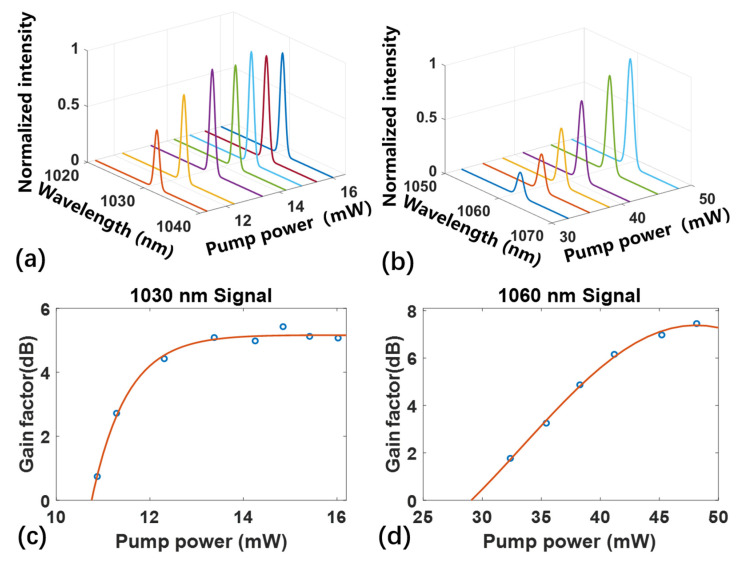
Gain characterization of the Yb^3+^-doped LN waveguides. Measured signal spectra as a function of increasing pump powers measured at (**a**) 1030 nm and (**b**) 1060 nm. Measured net internal gain as a function of increasing pump powers at different signal wavelength (**c**) 1030 nm and (**d**) 1060 nm.

**Figure 6 micromachines-13-00865-f006:**
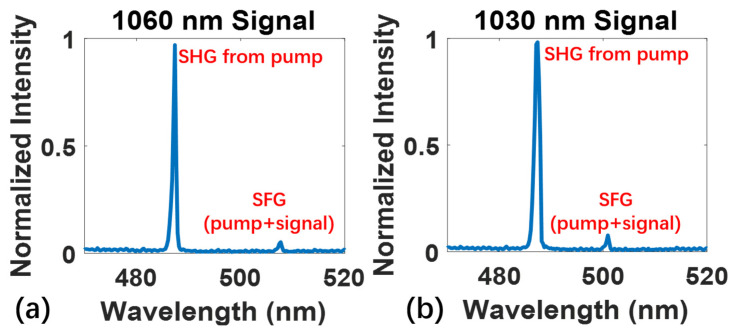
The spectral details of nonlinear frequency conversion from the signal light at (**a**) 1060 nm and (**b**) 1030 nm and the pump light at 976 nm.

## Data Availability

Data underlying the results presented in this Letter are not publicly available at this time but may be obtained from the authors upon reasonable request.

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
