# Peer review of "On-Chip Integrated Yb3+-Doped Waveguide Amplifiers on Thin Film Lithium Niobate"

_micromachines, 2022, doi:10.3390/mi13060865_

Round 1
Reviewer 1 Report
Title: On-chip integrated Yb3+-doped waveguide amplifiers on thin film lithium niobate
Authors: Fang, et al.
This work reports an Yb doped LNOI waveguide amplifier with gain of 1.25 dB/cm at 1030 nm and 2 dB/cm at 1060 nm. The results are very interesting and can benefit the research society of integrated waveguide amplifiers and lasers. I recommend to accept this work if the following questions are properly addressed.
1) The mode profiles in Fig.3b-3d seem not symmetric. Did the authors use bent waveguide in the simulation? If yes, what is the bending diameter?
2) The coupling loss between waveguide and pump is quite high. Can authors comment on the methods to improve the coupling efficiency?
3) The SHG means a loss for the signal at 1030 and 1060 nm. Will this become a power limitation on the increase of signal power if the device is not aimed for nonlinear process.
4) What is the overlap between the pump and signal? Please also give the definition of overlap that authors use.
Author Response
- The mode profiles in Fig.3b-3d seem not symmetric. Did the authors use bent waveguide in the simulation? If yes, what is the bending diameter?
Our response: In the article, the mode profiles in Fig.3b-d is the images of the waveguide output by an infrared camera and further processed by MATLAB. That is the reason why it looks like the simulation results. For the questions about asymmetric, the possible reason is that there are some angular differences in the coupling between the lensed fiber and the waveguide port under different wavelength conditions. The difference in the coupling angle causes slight distortions in the output modes of light at different wavelengths after being transmitted in the waveguide. Likewise, within the entire waveguide, there may be some machining errors that may increase or decrease the waveguide cross-section at certain locations, which will also lead to mode distortion. All of these reasons will cause the output mode profiles not symmetric. And the bending diameter of waveguides is 800nm. The large bending radius helps to avoid exciting higher-order modes in the waveguide and also has lower bending loss.
Action: this sentence has been added in line 149, as follows:
and the bending radius is 800 nm.
- The coupling loss between waveguide and pump is quite high. Can authors comment on the methods to improve the coupling efficiency?
Our response: In the article, the waveguide coupled with lensed fiber. Since the port area of the waveguide is smaller than the focusing spot size of lensed fiber, the mode field is not matched, so there will be a higher coupling loss.
There are many ways to reduce coupling loss as we know. First, fix the lensed fiber on the stage by using end caps. In order to reduce the impact of mechanical vibration of the stage, which has been used in experiments. Second, use the thermoelectric coolers (TEC) under the chip. When the pump light coupled in the waveguide, the temperature on the chip will change, which will cause the chip to deform. The deformation will shift the relative position between the lensed fiber and the waveguide port which will reduce the coupling efficiency. A TEC is used to control temperature stabilization on the chip in the experiment. And the method that can improve the coupling efficiency is to use a mode size converter. One end of the mode size converter has a large mode size that matches with the mode size of the standard fiber, the other end has a smaller mode size, which can be matched with the waveguide. This method will be used in subsequent experiments in the future.
Action: this sentence has been added in line 208, as follows:
using a mode size converter to improve the coupling efficiency.
- The SHG means a loss for the signal at 1030 and 1060 nm. Will this become a power limitation on the increase of signal power if the device is not aimed for nonlinear process.
Our response: In the article, we mentioned that the SHG signal measured at saturated pump power. Under this condition, the pump and signal power are high that SHG signal is obvious. When the laser intensity is very high, it can cause atomic polarization of the crystalline material. This is why SHG signals obvious under high power pump and signal light conditions. The generation of SHG consumes pump light and signal light, which will reduce the gain and limit the maximum output signal light power. In order to increase the final output signal light power of the waveguide amplifier, nonlinearity in the waveguide need be considered.
Action: this sentence has been added in line 194, as follows:
The SHG and SFG is harmful to the gain of waveguide amplifiers. Therefore, in order to achieve high power amplification, the nonlinear process need to be suppressed.
- What is the overlap between the pump and signal? Please also give the definition of overlap that authors use.
Our response: The overlap between the pump and signal means that the amplification process of ytterbium ions can be effectively inversed only in the region where the pump and signal act together in the waveguide. Due to the difference in wavelength between the pump and signal, their spot sizes after being focused by lensed fiber are different. Therefore, it is considered that only the overlapping part of the two light spots can be counted as the effective magnification area, which is the explanation of the overlap between the pump and signal.
Reviewer 2 Report
The authors report on the realization of the first Yb3+-doped amplifiers in thin film LN on insulator and give detailed report on it. Though, I believe, they published works on lasers in similar material these work has its own merits. There are few minor corrections:
1. The fabrication of the device should have more details even if these were published before- the authors should add a couple more sentences on it.
2. In the equation for the internal gain (line 156)- the formatting is not clear (a log of what base?) Also the term ??L is called differently on line 158-correct, please.
Author Response
Reviewer2:
The authors report on the realization of the first Yb3+-doped amplifiers in thin film LN on insulator and give detailed report on it. Though, I believe, they published works on lasers in similar material these work has its own merits. There are few minor corrections:
- The fabrication of the device should have more details even if these were published before- the authors should add a couple more sentences on it.
Our response: Thanks for the reviewer’s suggestion, we added the details in the line 76. The process flow include: (1) deposition of a thin layer of chromium (Cr) with a thickness of 500 nm on the surface of the LNOI by magnetron sputtering; (2) patterning of Cr film using femtosecond laser ablation to form the Cr mask; (3) etching of the LNOI layer by chemo-mechanical polishing to transfer the pattern of the Cr mask to the LN film; (4) removal of the Cr mask left on the surface of LNOI by chemical wet etching; (5) polishing again by chemo-mechanical polishing to smooth the surface of waveguide; (6) cleaning the LNOI waveguide to remove the dirty. Specific processing details such as laser parameters are mentioned in previous articles of our group.
Action: this sentence has been added in line 78, as follows:
The process flow is as follows: First, a femtosecond laser was used to ablate the Cr layer to process the waveguide pattern of Cr mask. Second, chemical mechanical polishing was used to etch the LNOI layer to transfer the waveguide pattern of the Cr mask to the LN film, then removed the Cr mask above the waveguide by chemical wet etching and carried on chemical mechanical polishing again. Finally, the waveguide after cleaning was ready for testing.
- In the equation for the internal gain (line 156)- the formatting is not clear (a log of what base?) Also the term ?/L is called differently on line 158-correct, please.
Our response: Thanks to the reviewer for carefully pointing out our mistake. The formula for the internal gain is a general formula for calculating decibels.
and are the output powers of the signal light with and without the excitation of the pump light, the ratio of two powers is the magnification of the output power. The logarithm of the ratio is converted to decibel units. This is the measured output total gain. is the optical propagation loss per unit length, it was measured in the experiment. L is the length of waveguide. The product of the two terms represents the total transmission loss for the entire waveguide. The internal gain of the waveguide is obtained by subtracting the total transmission loss from the total output gain.
Action: this sentence has been corrected, as follows:
And is the optical propagation loss per unit length.

Reviewer 3 Report
The authors characterize the optical gain of ytterbium-doped thin film lithium niobate waveguide amplifiers. To the authors’ knowledge, ytterbium-doped thin film lithium niobate waveguide amplifiers have not been demonstrated before. They measure an internal net gain of 5 dB at 1030 nm and 8 dB at 1060 nm in a 4.0 cm long waveguide pumped by a 976 nm laser diode. These results add another useful functionality to the thin film niobate platform with its low propagation loss and large optical nonlinearities.
The addition of gain to the thin film lithium platform is definitely of significant interest to the photonics community. Yet, some of the results presented by the authors require further clarification and explanation to ensure a technically sound and complete manuscript with results that can be understood and reproduced by others in the community. Below are my comments/questions for the authors.
· Line 51: Please clarify briefly why it is challenging to achieve high gain in ion-diffused waveguides.
· Introduction: Also mention the recent efforts to integrate III-V gain media with lithium niobate, e.g. https://opg.optica.org/optica/fulltext.cfm?uri=optica-8-10-1288&id=460068 and https://opg.optica.org/optica/fulltext.cfm?uri=optica-9-4-408&id=471161
· Materials and methods: Explain how you dope the lithium niobate with ytterbium. Is the doping done before or after waveguide patterning? Which parameters do you use for ion implantation? How do you ensure the ions are at the correct depth?
· Materials and methods: What is the thickness of the silica bottom cladding? Which substrate material do you use? Silicon, lithium niobate, other?
· Line 84: “so that the single-mode transmission is mainly used in the waveguide”. Why is the word “mainly” used here? Is there some uncertainty as to whether the waveguides are single mode or not?
· Line 85: Provide a reference for the energy level diagram.
· Figure 2c: Is this a map of patterned waveguides? Then why don't we see waveguides in the figure? Is the silicon dioxide bottom cladding also doped with ytterbium?
· Materials and methods: Are you using single-frequency laser diodes? What are their linewidths? What are the linewidths of the ytterbium transitions?
· Line 105: “The polarization states of both the pump and signal laser beam are adjusted using in-line fiber polarization controllers to achieve the best amplification performance.” Can you elaborate on the difference in performance for TE and TM waveguide modes? Is there a significant difference in gain between both, if yes, why? Which polarization are you using in the measurements?
· Figure 3b,c,d: Insert scale bars.
· Line 141: “the propagation loss of the pump light at 976 nm is 6.78 dB/cm”. Is this the expected propagation loss for the known doping concentration?
· Line 142: Compare the measured propagation losses to literature.
· Line 148: “the lensed fiber is multi-mode”. Do you notice any changes in coupling loss when moving the fiber around, due to changes in power distribution between the modes in the fiber?
· Line 160: “The full width at half maximum of two narrow signal peaks are about 1.5 nm.” Explain why the peaks are this wide. Is this due to limited resolution of the spectrometer, due to amplified spontaneous emission, other factors?
· Results and discussion: Are the measured gain and saturation powers as expected? Compare with literature.
· Line 169: What is limiting the efficiency to 5%?
· Results and discussion: What signal powers are you using in your gain measurements? High signal powers will cause saturation. The authors should measure the gain for different signal powers and also plot the gain vs signal power (for different pump powers).
· Results and discussion: For the gain measurements, how do you remove the effect of amplified spontaneous emission (ASE) from the gain measurement? A significant amount of ASE can distort the calculated gain profile if not taken into account.
· Line 194: “Further optimizations concerning the Yb3+ ions doping concentration to increase the absorption for pump light, and the waveguide geometric design to allow for high-power amplifications, are anticipated to increase the waveguide gain to even higher values, showing the potential of the on-chip applications for photonic integrated circuit (PIC).” Can the authors be more specific and provide numbers where possible?
Author Response
Reviewer3:
The authors characterize the optical gain of ytterbium-doped thin film lithium niobate waveguide amplifiers. To the authors’ knowledge, ytterbium-doped thin film lithium niobate waveguide amplifiers have not been demonstrated before. They measure an internal net gain of 5 dB at 1030 nm and 8 dB at 1060 nm in a 4.0 cm long waveguide pumped by a 976 nm laser diode. These results add another useful functionality to the thin film niobate platform with its low propagation loss and large optical nonlinearities.
The addition of gain to the thin film lithium platform is definitely of significant interest to the photonics community. Yet, some of the results presented by the authors require further clarification and explanation to ensure a technically sound and complete manuscript with results that can be understood and reproduced by others in the community. Below are my comments/questions for the authors.
- Line 51: Please clarify briefly why it is challenging to achieve high gain in ion-diffused waveguides.
Our response: The ytterbium-doped lithium niobate waveguide mentioned above is processed by ion diffusion technology. The difference in refractive index between the inside and outside of the waveguide is small. Therefore, its constraint ability of light is poor. The waveguide amplifier needs a long enough propagation distance to absorb the pump light efficiently, which requires a high constraint ability. That is why we say it is challenging to achieve high gain in ion-diffused waveguides.
- Introduction: Also mention the recent efforts to integrate III-V gain media with lithium niobate,
e.g. https://opg.optica.org/optica/fulltext.cfm?uri=optica-8-10-1288&id=460068 and https://opg.optica.org/optica/fulltext.cfm?uri=optica-9-4-408&id=471161
Our response: Thanks for the reviewer’s suggestion, we added the recent efforts in the line 76.
Action: this sentence has been added in line 48, as follows:
It has been demonstrated integrated high-power lasers on TFLN platform [16,17].
- Materials and methods: Explain how you dope the lithium niobate with ytterbium. Is the doping done before or after waveguide patterning? Which parameters do you use for ion implantation? How do you ensure the ions are at the correct depth?
Our response: The Yb3+ doped lithium niobate wafers are produced by Jinan Jingzheng Electronics Co., Ltd. As we ask, the company uses the Czochralski method to grow doped lithium niobate crystals. In the method, the raw material of the crystal to be grown is heated and melted in a high temperature-resistant crucible, a seed crystal is placed on the seed rod, and the seed crystal contacts the surface of the melt. After the surface of the seed crystal is slightly melted, the seed crystal is pulled and rotated. The rod makes the melt in a supercooled state and crystallizes on the seed crystal. In the process of continuous pulling and rotation, the doped crystal to be processed is grown. Then the grown ytterbium-doped lithium niobate crystal is processed into wafers. Since we purchased ytterbium-doped lithium niobate wafers directly, the doping method and specific parameters are not convenient to write in the article. And the doping done before waveguide patterning. The advantage of the ytterbium-doped lithium niobate wafer fabricated by the Czochralski method is that the ytterbium ions are uniformly distributed in the lithium niobate layer. The problem of uneven distribution with depth of the ion implantation method does not exist.
- Materials and methods: What is the thickness of the silica bottom cladding? Which substrate material do you use? Silicon, lithium niobate, other?
Our response: The structure of ytterbium-doped lithium niobate wafer is divided into four layers, the surface layer is 500 nm thick Cr, the second layer is 600 nm thick ytterbium-doped lithium niobate, and the third layer is 2 mm silicon dioxide layer. The bottom layer is a 0.5 mm thick silica substrate. we added the details in the line 75.
Action: this sentence has been added, as follows:
The structure of ytterbium-doped lithium niobate wafer as follows from top to bottom is 500 nm thick Cr layer, 600 nm thick ytterbium-doped lithium niobate layer, 2 m silicon dioxide layer and 0.5 mm thick silica layer.
- Line 84: “so that the single-mode transmission is mainly used in the waveguide”. Why is the word “mainly” used here? Is there some uncertainty as to whether the waveguides are single mode or not?
Our response: The pump light and the signal light are focused and coupled into the waveguide by lensed fiber, both single-mode and multi-mode are coupled into the waveguide. The multimode will gradually scatter out of the waveguide during transmission. So, the output mode of waveguide is only single mode. This is the reason that the single-mode transmission is mainly used in the waveguide.
- Line 85: Provide a reference for the energy level diagram.
Our response: Thanks for the reviewer’s suggestion, we added the reference in the line 291.
Action: the reference has been added, as follows:
- E. Montoya, A. Lorenzo, and L. E. Bausa, “Optical performance of Yb3+ in LiNbO3 laser crystal,” J. Phys.: Condens. Matter. 1999,11, 311–320.
- Figure 2c: Is this a map of patterned waveguides? Then why don't we see waveguides in the figure? Is the silicon dioxide bottom cladding also doped with ytterbium?
Our response: Figure 2c is an EDS map of Yb3+-doped LNOI wafer, which is used for processing waveguides. The sampling depth of EDS is about 1 μm, the sampling area in the wafer will be damaged by X ray, so the waveguides are not placed in the map area in order to protect waveguide. We selected the area near the waveguide for sampling, so there is no waveguides in the figure. The silicon dioxide layer is not doped with ytterbium.
Action: The misunderstanding word has revised in the sentence, as follows:
Figures 2(b) and (c) showing situ quantitative elements analysis of fabricated Yb3+-doped wafer measured by energy dispersive spectrometer (EDS) (Zeiss Gemini SEM450)
- Materials and methods: Are you using single-frequency laser diodes? What are their linewidths? What are the linewidths of the ytterbium transitions?
Our response: All the laser diodes are single frequency. The linewidth of 1030 nm and 1060 nm laser diodes is 1.5 nm, and the linewidth of 980 nm laser diode is 3 nm. The linewidth of the ytterbium transitions is about 1 nm.
Action: the detail has been added in line 112, as follows:
Here, a pump laser at 976 nm is provided by a single frequency laser diode (CM97-1000-76PM, II-VI Laser Inc.), while single frequency laser diodes at wavelength 1030 nm and 1060 nm (II-VI Laser Inc.) are used as the signal light.
- Line 105: “The polarization states of both the pump and signal laser beam are adjusted using in-line fiber polarization controllers to achieve the best amplification performance.” Can you elaborate on the difference in performance for TE and TM waveguide modes? Is there a significant difference in gain between both, if yes, why? Which polarization are you using in the measurements?
Our response: In the experiment we used in-line fiber polarization controllers to control the polarization states of the pump and signal laser. The TE and TM modes behave differently in waveguide. According to reference 25, the gain is the highest when both the pump light and the signal light are in TE mode, but there is almost no amplification when both are in the TM mode. According to this conclusion, we set the pump light and signal light modes in the TE mode in the experimental design, in order to measure the highest gain.
Action: the detail has been added in line 115, as follows:
The polarization states of both the pump and signal laser beam are set to the TE modes by using in-line fiber polarization controllers to achieve the best amplification performance.
- Figure 3b,c,d: Insert scale bars.
Our response: Thanks for the reviewer’s suggestion, we added the scale bar in the Figure3. The scale bar of three figures is the same.
Action: we added the scale bars in Fig 3, as follows:
- Line 141: “the propagation loss of the pump light at 976 nm is 6.78 dB/cm”. Is this the expected propagation loss for the known doping concentration?
Our response: The propagation loss of the pump light is the best result after optimization in the experiment. The doping concentration affects the scattering and absorption losses caused by the waveguide material. The absorption cross section of ytterbium ions at 980 nm is very large, so the higher the concentration of ytterbium ions, the greater the absorption loss. The main reason for the scattering loss is the uneven distribution of the medium inside the waveguide, which is called Rayleigh scattering. Rayleigh scattering is proportional to . Loss due to Rayleigh scattering decreases as wavelength increases and Rayleigh scattering is independent of doping concentration. And there are many factors that affect propagation loss, such as sidewall roughness, bending loss, etc. To sum up, it cannot be determined whether the transmission loss is expected at the known doping concentration. We can only keep the transmission loss as low as possible in our experiments.
- Line 142: Compare the measured propagation losses to literature.
Our response: Since there is no ytterbium-doped waveguide amplifier literature, we compare with erbium-doped waveguide amplifiers. the propagation loss of the pump light in EDWA at 976 nm is 6.77 dB/cm and the signal light is 0.16 dB/cm [24], close to our experimental results. Compared to the general ytterbium-doped fiber amplifiers, the propagation loss in YDWA is 2-3 orders magnitude higher.
- Line 148: “the lensed fiber is multi-mode”. Do you notice any changes in coupling loss when moving the fiber around, due to changes in power distribution between the modes in the fiber?
Our response: There is an optimal coupling point for lensed fiber coupled with waveguides where coupling loss is the lowest. When moving the lensed fiber around, the coupling loss will increase. The main reason is due to the offset of the coupling position between the fiber lens and the waveguide port. The increase in coupling loss due to mode changes in the fiber is not the main reason. And coupling loss also increased when the lensed fiber was bent. So, we fixed the lensed fiber to avoid vibration and make sure it was straight in the experiment to reduce the effect on coupling loss.
- Line 160: “The full width at half maximum of two narrow signal peaks are about 1.5 nm.” Explain why the peaks are this wide. Is this due to limited resolution of the spectrometer, due to amplified spontaneous emission, other factors?
Our response: The full width at half maximum of signal peaks is follow the linewidth of single frequency laser diode. Because the properties of the photons emitted by the stimulated radiation are consistent with the incident signal photons, the properties of the incident photons are determined by the signal light laser diode.
Action: the detail has been added in line 171, as follows:
The full width at half maximum of two narrow signal peaks are about 1.5 nm, it follows the linewidth of laser diodes.
- Results and discussion: Are the measured gain and saturation powers as expected? Compare with literature.
Our response: This result is the best result we have measured in experiments. Since the Yb3+-doped lithium niobate waveguide is the first time that our group has processed, there are currently no other articles on Yb3+-doped lithium niobate waveguide amplifiers to compare. Erbium-doped lithium niobate waveguide amplifiers can now achieve 20dB [28], III-V heterogeneous integrated lithium niobate amplifiers achieve 11.8dB [Optica 8, 1288 (2021)]. Compared with these lithium niobate amplifiers, the gain of our ytterbium-doped waveguide amplifier is not very high. The possible reason is that the doping concentration of ytterbium ions is relatively low. We will continue to optimize the ytterbium-doped waveguide in the future.
- Line 169: What is limiting the efficiency to 5%?
Our response: Quantum efficiency means the efficiency which pump light converted into signal light. The factors, such as ytterbium ion concentration, roughness of waveguide sidewalls, waveguide shape, waveguide uniformity along the entire transmission length, will limit the gain and efficiency.
- Results and discussion: What signal powers are you using in your gain measurements? High signal powers will cause saturation. The authors should measure the gain for different signal powers and also plot the gain vs signal power (for different pump powers).
Our response: The signal powers using in measurements is 8 microwatts. Since the power corresponding to the lowest intensity signal light spectrum that the spectrometer used in the experiment can resolve is in the order of microwatt, 8 microwatts was selected as a more suitable power while ensuring the stability of the signal spectrum. The experiment is limited by the spectrum acquisition equipment. If a higher resolution spectrometer can be used instead, the signal light with lower power can be measured, and the gain measurement can be performed better.
- Results and discussion: For the gain measurements, how do you remove the effect of amplified spontaneous emission (ASE) from the gain measurement? A significant amount of ASE can distort the calculated gain profile if not taken into account.
Our response: We did not measure the amount of ASE in our experiments, and ASE is inevitable quantum noise. The measured gain was the final output gain from the waveguide. the final output gain has already affected by ASE during propagation. Although the specific amount of ASE is unknown, the final measured gain is accurate. In the fiber, the influence of ASE will be obvious only when the gain is high (>30 dB). But the gain is not very high in YDWA, the influence of ASE will be smaller than other factors.
- Line 194: “Further optimizations concerning the Yb3+ ions doping concentration to increase the absorption for pump light, and the waveguide geometric design to allow for high-power amplifications, are anticipated to increase the waveguide gain to even higher values, showing the potential of the on-chip applications for photonic integrated circuit (PIC).” Can the authors be more specific and provide numbers where possible?
Our response: Yb3+ comprises a simple two-level system and provides efficient lasing around the 1 μm window. The exact Yb3+ sub-level splitting depends on the glass composition and Yb3+ concentration [J. Non-Crystalline Solids, vol. 354, pp. 4760–4764, 2008]. And with the increase of Yb3+ from 0.5 to 5 mol%, the absorption across section decrease to less than 8%, the measured lifetime of 2F5/2 decrease from 1.9 to 1.25 ms in silica glasses [Optics Communications 253 (2005) 151–155]. From these conclusions, it can be seen that changing the doping concentration has an effect on the gain of the waveguide amplifier. Therefore, the doping concentration of ytterbium ions in the lithium niobate substrate is an optimal condition. The specific doping concentration can be beneficial to the gain still needs to be measured in the experiment, and the specific value cannot be given temporarily. About the optimization of the waveguide geometric design, it can be optimized from two directions: optimizing the shape of the waveguide cross-section and using the mode spot converter. The waveguide shape used in the present article is a rib waveguide, which can be optimized by using a ridge waveguide configuration. Using a mode-spot converter can reduce the coupling loss, thereby increasing the power of the pump light coupled into the waveguide, which can excite more ytterbium ions to the excited state, resulting in an increase in the gain of the waveguide amplifier. The above methods are the specific details of the subsequent optimization of the ytterbium-doped waveguide amplifier.
